# Validation of the Korean version of the composite autonomic symptom scale 31 in patients with Parkinson's disease

Jong Hyeon Ahn[1,2☯], Jin Myoung Seok[3☯], Jongkyu Park[3], Heejeong Jeong[4], Younsoo Kim[5], Joomee Song[1,2], Inyoung Choi[1,2], Jin Whan Cho[1,2], Ju-Hong Min[1,2], Byoung Joon Kim[1,2]*, Jinyoung Youn[1,2]*

1 Department of Neurology, Samsung Medical Centre, Sungkyunkwan University School of Medicine, Seoul, Republic of Korea, 2 Neuroscience Centre, Samsung Medical Centre, Seoul, Republic of Korea, 3 Department of Neurology, Soonchunhyang University Cheonan Hospital, Soonchunhyang University College of Medicine, Cheonan, Republic of Korea, 4 Department of Neurology, Gyeongsang National University Changwon Hospital, Changwon, Republic of Korea, 5 Department of Neurology, Samsung Changwon Hospital, Sungkyunkwan University School of Medicine, Changwon, Republic of Korea

☯ These authors contributed equally to this work.
* bjkim@skku.edu (BJK); genian@skku.edu (JY)

**Data Availability Statement:** All relevant data are within the paper and its Supporting information files.

**Funding:** The author(s) received no specific funding for this work.

## Abstract

### Purpose

The composite autonomic symptom scale-31 (COMPASS-31) is a self-rated questionnaire that evaluates diverse autonomic symptoms. In the present study, we developed the Korean version of the COMPASS-31 (K-COMPASS-31) with appropriate translation, and verified its reliability and internal and external validity in patients with Parkinson's disease (PD).

### Methods

The original COMPASS-31 was translated independently into Korean by two bilingual neurologists. Test-retest reliability was evaluated at a 2-week interval. We investigated the correlations between the K-COMPASS-31, the scale for outcomes in PD-autonomic (SCOPA-AUT), and the results of an autonomic function test (AFT), respectively.

### Results

A total of 90 patients with PD (47 females; mean age, 63.4 ± 10.8 years) were enrolled. The K-COMPASS-31 showed excellent test-retest reliability (intra-class correlation coefficient = 0.874, $p < 0.001$) and internal validity (Cronbach's α-coefficient = 0.878). The COMPASS-31 was positively correlated with SCOPA-AUT ($r = 0.609$, $p < 0.001$) and the results of the AFT.

### Conclusions

In conclusion, the K-COMPASS-31 showed excellent reliability and validity for the assessment of autonomic symptoms in PD patients. The K-COMPASS-31 is an easy-to-repeat and widely used tool for investigating autonomic dysfunction in various neurologic disorders

**Competing interests:** The authors have read the journal's policy and have the following competing interests: JY received speakers' honoraria from SK Chemicals and Boston Scientific, and research support from Medtronic and Boston Scientific. However, this support is unrelated to this study. This does not alter our adherence to PLOS ONE policies on sharing data and materials. There are no patents, products in development or marketed products associated with this research to declare.

and enables comparison of autonomic dysfunction among neurologic disorders. We recommend the K-COMPASS-31 as a valid instrument for use in clinical practice for patients with PD.

## Introduction

Autonomic dysfunction is one of the central features of non-motor symptoms in Parkinson's disease (PD) [1]. Objective evaluation of autonomic function in PD is an essential part of differential diagnosis [2]; however, autonomic function tests require extensive time and expensive equipment only available at tertiary centres. Thus, some studies have used patient-reported questionnaires as a surrogate for the assessment of autonomic dysfunction in PD patients [3]. The scale for outcomes in Parkinson's disease-autonomic (SCOPA-AUT), which was developed to evaluate autonomic symptoms, is widely used in patients with PD [4]. To better understand autonomic dysfunction in PD in comparison with other neurological disorders that involve the autonomic system, other reliable questionnaires that include parameters that can be compared with the results of autonomic function testing (AFT) are needed.

The composite autonomic symptom scale-31 (COMPASS-31) questionnaire, a self-rated questionnaire that evaluates diverse autonomic symptoms [5], has been reported as a useful tool for the evaluation of autonomic dysfunction in many neurological diseases including multiple sclerosis, small fibre neuropathy, and parkinsonism [6–8]. Therefore, in the present study, we developed a Korean version of the COMPASS-31 with appropriate translation, and verified its reliability and internal and external validity in patients with PD by evaluating agreement between the COMPASS-31 and AFT results.

## Materials and methods

### Subjects and clinical assessments

This study was approved by the local institutional review board of Samsung Medical Centre, Soonchunhyang University Hospital Cheonan, Gyeongsang National University Changwon Hospital, and Samsung Changwon Hospital, and all enrolled subjects provided written informed consent. We prospectively enrolled patients with PD from January 2019 to October 2020 from movement disorder clinics at four tertiary medical centres including Samsung Medical Centre, Soonchunhyang University Hospital Cheonan, Gyeongsang National University Changwon Hospital, and Samsung Changwon Hospital. Patients were enrolled if they were diagnosed with PD based on the UK Brain Bank Criteria for PD [9]. Patients were excluded if they had structural brain lesions, other known neurodegenerative diseases, cognitive impairment (mini-mental status examination score of < 26 or fulfilment of *DSM-IV* criteria for dementia) [10], psychiatric disorders requiring medication, malignancy, or musculo-skeletal problems mimicking parkinsonism. In addition, we also excluded patients with medical conditions including cardiac failure or arrhythmia, end-stage renal disease, diabetes mellitus, and other autonomic neuropathies that could affect AFT results.

Demographic and clinical data were collected for all enrolled patients. Parkinsonian motor symptoms were evaluated with the Unified Parkinson's Disease Rating Scales (UPDRS) part III, and modified Hoehn and Yahr (H&Y) stages during the medication 'ON' state [11,12]. The UPDRS part III was grouped into tremor (items 20 and 21), rigidity (item 22), bradykinesia (items 23, 24, 25, 26, and 31), and axial motor symptoms (items 27, 28, 29, and 30) [13].

Patients were classified into three subtypes according to dominant parkinsonian symptoms: tremor-dominant (TD), akinetic-rigid (AR) and the mixed subtypes [14]. Levodopa equivalent daily dose (LEDD) was calculated based on a previous study [15]. Global cognition was checked with the Korean version of the mini-mental state examination (K-MMSE) [16] and parkinsonian non-motor symptoms were evaluated with the Korean version of the non-motor symptoms scale for PD (K-NMSS) [17]. In addition, quality of life was assessed using PD questionnaire-39 (PDQ-39) [18].

## COMPASS-31 questionnaire and translation

The COMPASS-31 questionnaire consists of 31 items in 6 domains: orthostatic intolerance (4 items), vasomotor (3 items), secretomotor (4 items), gastrointestinal (12 items), bladder (3 items), and pupillomotor (5 items). The total sum score for all domains adjusted with each weighting factor is from 0 to 100; a higher COMPASS-31 score indicates more severe autonomic symptoms [5]. The original COMPASS-31 English version was translated into Korean independently by two bilingual neurologists, who subsequently worked together to create a single Korean version. A panel of authors reviewed discrepancies between the Korean translation version and the original English version to confirm the accuracy of the Korean version of the COMPASS-31 (K-COMPASS-31). Finally, we tested the K-COMPASS-31 in 5 patients with PD, and interviews with them were conducted for validation.

## Validation procedure

The K-COMPASS-31 was completed twice at 2-week intervals; the second K-COMPASS-31 score was used only for test-retest reliability evaluation. In addition to the K-COMPASS-31, the SCOPA-AUT and AFT battery were also applied in all patients as comparators for validation.

SCOPA-AUT is a brief, widely used questionnaire for patients with PD, which has 25 items that assess six areas; the item scores range from 0 ('never') to 3 ('often') with a maximum score of 69. The Korean version of the SCOPA-AUT was validated recently [19]. Our AFT battery included heart rate response to deep breathing (HRDB), the Valsalva manoeuvre, sympathetic skin response (SSR), and blood pressure and heart rate response to the head-up tilt test (HUT). Enrolled patients underwent the AFT battery in a standardized manner as previously reported [19–21].

## Statistical analyses

Appropriate summary statistics were used to describe categorical and continuous variables. Continuous data were presented as mean with standard deviation (SD) or median with interquartile range (IQR); categorical variables were presented as absolute and relative frequencies. Sample size for the reliability test was calculated using the analysis methods suggested by Walter *et al.* with an acceptable reliability of 0.75, an expected reliability of 0.9, a significant level of 0.5 and a power of 80% [22]. Cronbach's α-coefficient was used for internal consistency analysis of the K-COMPASS-31 total and for each domain, and test-retest reliability was assessed using the intra-class correlation coefficient (ICC). Validity was assessed using correlation analysis between the K-COMPASS-31 and SCOPA-AUT; Pearson partial correlation was adjusted for potential confounders that included age, sex, disease duration, education year, PD subtype, UPDRS III, and LEDD. The association of the K-COMPASS-31 with other variables including age, disease duration, UPDRS, H&Y, LEDD, NMSS, and PDQ-39 summary index (PDQ-39 SI) was also investigated. A $p$-value $< 0.05$ was considered significant. Statistical analyses were performed using SPSS for Windows version 25 (SPSS Inc. Version 25.0 Chicago, IL).

## Results

### Subjects and clinical characteristics

A total of 90 patients with PD (47 females, 52.2%; mean age, 63.4 ± 10.8 years) were finally enrolled. The mean disease duration was 4.2 ± 4.2 years, H&Y was 1.8 ± 0.8, and UPDRS part III score was 19.6 ± 11.3; LEDD was 417.7 ± 346.9 and only one patient was drug-naïve. Twenty-four (26.7%) patients exhibited the TD subtype, 58 patients (64.4%) the AR, and 8 patients (8.9%) the mixed subtype. Autonomic symptoms and AFT results were evaluated in all enrolled patients, and mean SCOPA-AUT score was 16.4 ± 11.2. The results of the AFT battery including HRDB, the Valsalva manoeuvre, blood pressure, and heart rate response to the HUT, sympathetic skin response and orthostatic hypotension (OH) are presented in Table 1.

### Intra-individual reliability of the K-COMPASS-31

The total K-COMPASS-31 score for enrolled patients with PD was 22.0 ± 17.4 and ranged from 0 to 75.8; the K-COMPASS-31 scores for each of the six domains of autonomic symptoms are presented in Table 2. Thirty-three patients in our study underwent K-COMPASS-31 test-retest, which showed excellent test-retest reliability with an intra-class correlation coefficient was 0.874 (95% CI, 0.744–0.938; p < 0.001). Cronbach's α-coefficient for the K-COMPASS-31 was 0.878, and the six domains of the K-COMPASS-31 also showed good internal validity (Table 2).

### Correlation with other objective and subjective measurements

Total and subdomain scores for the K-COMPASS-31 were compared with clinical features, AFT results, and the SCOPA-AUT for validation. There were significant correlations between K-COMPASS-31 score and the AFT results. E:I ratio and Valsalva ratio were negatively

**Table 1. Demographic data and results of autonomic function tests for enrolled patients with Parkinson's disease.**

|  | Enrolled patients with PD (n = 90) |
|---|---|
| Female, n (%) | 47 (52.2) |
| Age, years (SD) | 63.4 (10.8) |
| Disease duration, years (SD) | 4.2 (4.2) |
| UPDRS part III (SD) | 19.6 (11.3) |
| Motor subtype, TD/mixed/AR, n (%) | 24/8/58 (26.7/8.9/64.4) |
| H & Y stage (SD) | 1.8 (0.8) |
| LEDD, mg (SD) | 417.7 (346.9) |
| MMSE (SD) | 27.3 (2.1) |
| *Autonomic function profile* |  |
| SCOPA-AUT (SD) | 16.4 (11.2) |
| E:I ratio (IQR) | 1.11 (1.07–1.18) |
| Valsalva ratio (IQR) | 1.35 (1.18–1.53) |
| Pressure recovery time, sec (IQR) | 1.7 (1.0–4.7) |
| Abnormality in SSR, n (%) | 13 (14.4) |
| Orthostatic hypotension, n (%) | 23 (25.6) |

PD, Parkinson's disease; SD, Standard deviation; IQR, Interquartile range; UPDRS, Unified Parkinson's disease rating scale; TD, Tremor dominant subtype; AR, Akinetic-rigid subtype; H & Y, Modified Hoehn and Yahr; LEDD, Levodopa equivalent daily dose; MMSE, Mini-mental status exam; SCOPA-AUT, Scale for outcomes in Parkinson's disease-autonomic; E:I, Expiratory: Inspiratory.

**Table 2. Total and domain scores of the K-COMPASS-31.**

| K-COMPASS-31 domains | Total (n = 90) | | Cronbach's α |
| --- | --- | --- | --- |
| | Mean (SD) | Median (range) | |
| Total score | 22.0 (17.4) | 17.4 (0–75.8) | 0.878 |
| Orthostatic intolerance | 9.7 (11.3) | 4.0 (0–36.0) | 0.879 |
| Vasomotor | 0.3 (0.9) | 0 (0–4.2) | 0.906 |
| Secretomotor | 4.5 (3.6) | 4.3 (0–15.0) | 0.605 |
| Gastrointestinal | 5.0 (3.4) | 4.9 (0–13.4) | 0.679 |
| Bladder | 1.8 (2.1) | 1.1 (0–10.0) | 0.702 |
| Pupillomotor | 0.7 (1.0) | 0 (0–5.0) | 0.849 |

K-COMPASS-31, Korean version of the composite autonomic symptom score-31; SD, standard deviation.

correlated (r = -0.240, p = 0.023; r = -0.247, p = 0.019) indicating cardiovagal dysfunction [23], and pressure recovery time (PRT), which represented adrenergic function, was positively correlated with K-COMPASS-31 score (r = 0.345, p < 0.001) (Fig 1). However, after adjusting for confounders including age, sex, disease duration, education year, PD subtype, UPDRS III, and LEDD, the K-COMPASS-31 score was correlated with E:I ratio and PRT (Table 3).

The K-COMPASS-31 total scores were positively correlated with the SCOPA-AUT scores (r = 0.626, p < 0.001). Subdomain scores of the K-COMPASS-31 were also correlated with the SCOPA-AUT total score. The pupillomotor domain of the K-COMPASS-31 was not correlated with the total SCOPA-AUT but was correlated with the pupillomotor subscore of the SCOPA-AUT (Fig 1).

## Correlation with other clinical features of Parkinson's disease

K-COMPASS-31 total scores were correlated with disease duration (r = 0.328, p < 0.002). Among the UPDRS part III subscores, K-COMPASS-31 total score and subdomain scores were associated with the subscore of axial symptoms. There were positive correlations between K-COMPASS-31 total score and patient clinical features including LEDD, NMSS, and the PDQ-39 summary index (PDQ-39 SI) (Table 3).

## Discussion

This study is the first to validate a Korean language version of the COMPASS-31. The COMPASS-31 is an abbreviated questionnaire based on a well-established autonomic symptom profile [5]. This questionnaire has been translated into many languages and validated for various neurological diseases [6,8,24–27]. Here, we validated the Korean version of the COMPASS-31; the results of our study showed good reliability and internal validity for the K-COMPASS-31. Our study validated the COMPASS-31 translated into Korean, and includes the first validation study of the COMPASS-31 in patients with PD.

In PD patients, the SCOPA-AUT is a widely used questionnaire for the assessment of dysautonomia. The strength of the SCOPA-AUT is that it is a well-organized questionnaire specific to PD, and the SCOPA-AUT has been suggested to represent autonomic involvement in PD even better than the objective AFT [28]. At the same time, the limitation is that SCOPA-AUT is too focused on PD, and there was no significant correlation between SCOPA-AUT and AFT in a previous study [28]. Therefore, even though SCOPA-AUT is a sensitive measurement tool for autonomic dysfunction in PD, it is not especially useful for evaluating and comparing dysautonomia between PD and other diseases. However, the K-COMPASS-31 was

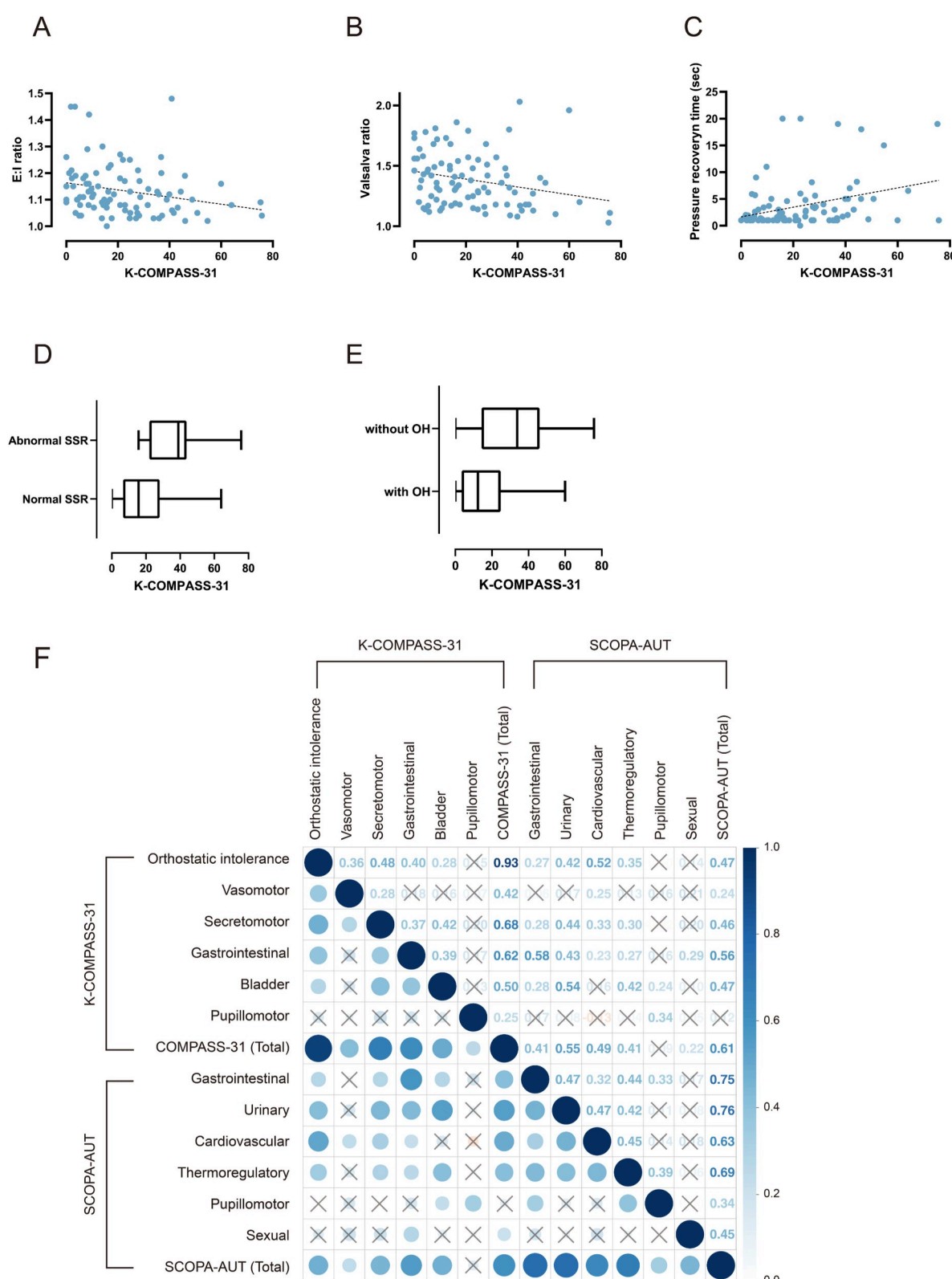

**Fig 1. Correlation between the K-COMPASS-31 and SCOPA-AUT, and parameters of the autonomic function test.** The K-COMPASS-31 correlated negatively with E:I ratio (r = -261, p = 0.017) (A), but not with Valsalva ratio (r = - 0.187, p = 0.091) (B), and correlated positively with pressure recovery time (r = 0.243, p = 0.027) (C). The K-COMPASS-31 score was higher in PD patients with abnormal

sympathetic skin response (SSR) (19.1 ± 15.4 vs. 39.0 ± 19.3, p < 0.001) (D) and with orthostatic hypotension (33.4 ± 21.4 vs. 18.1 ± 13.9, p < 0.001) (E). Correlation matrix revealed a positive correlation between the K-COMPASS-31 and SCOPA-AUT and their subdomain scores. Pearson's correlation coefficient values are shown (F). Correlations with nonsignificant p-values are denoted with an "X" (p ≥ 0.05).

significantly correlated with both the SCOPA-AUT and objective parameters of AFT in our study. Thus, the results of our study suggest that the K-COMPASS-31 could be a potential surrogate for evaluating autonomic function in patients with PD as well as other patients with various disorders that present with autonomic dysfunction.

Considering that the prevalence of autonomic dysfunction in PD patients has continuously increased over time, repetitive evaluation of autonomic symptoms may be needed, and the COMPASS-31 is more advantageous than the objective AFT for this in several aspects. First of all, the COMPASS-31 takes less time and is less costly. Additionally, autonomic dysfunction in PD patients involves various body systems that include gastrointestinal, urinary, sexual, cardiovascular, pupillary motor, and thermoregulatory function. Conventional AFT does not cover all types of autonomic dysfunction in PD patients, such as gastrointestinal, urinary, and pupillary dysfunction [29], which require specialized tests and equipment. Self-reported questionnaires like the COMPASS-31 include questions on various autonomic symptoms and can thus detect a variety of autonomic symptoms in PD patients. Therefore, validation of the K-COMPASS-31 can help to distinguish PD from multiple system atrophy and contribute to timely diagnosis and management of autonomic dysfunction to improve prognosis and quality of life for PD patients.

Autonomic dysfunction is a common and disabling symptom in PD patients. It increases the risk of falls, exacerbates motor dysfunction, and decreases quality of life [30]. Among types of autonomic dysfunction, OH affects nearly half of PD patients, representing a major non-motor symptom burden of PD [3,31]. In this study, 23 patients (25.6%) showed OH during HUT. The patients with OH had a higher total K-COMPASS-31 score and orthostatic intolerance subscore than those without OH (33.4 ± 21.4 *vs.* 18.1 ± 13.9, *p* < 0.001; 16.5±13.1 *vs.* 7.3 ± 9.6, *p* = 0.001). The total score and orthostatic intolerance subscore of the K-COMPASS-31 correlated moderately with PRT (r = 0.243; r = 0.220), which reflects adrenergic dysfunction and is defined as the time interval from the time of lowest blood pressure in phase 3 to when the blood pressure reaches baseline during the Valsalva manoeuvre [23,32]. Considering this difference in K-COMPASS-31 scores according to the presence of OH and the correlation between PRT and the K-COMPASS-31 scores, the K-COMPASS-31 score, especially its orthostatic intolerance subscore, might be helpful for evaluating characteristics of OH in patients with PD. For the evaluation of cardiovagal dysfunction, K-COMPASS-31 showed a negative correlation with E:I ratio, but was not correlated with Valsalva ratio after adjusting for confounders. A previous validation study of the COMPASS-31 showed similar results, in that COMPASS-31 correlated well with overall AFT score but not with AFT scores for cardiovagal function [26]. E:I ratio might be more suitable for evaluating cardiovagal dysfunction in patients with PD because of difficulties in performing the Valsalva manoeuvre [33].

In terms of motor symptoms, the K-COMPASS-31 was not correlated with UPDRS III total score but was correlated with the axial symptoms subscore. K-COMPASS-31 score was also associated with LEDD. We evaluated UPDRS III scores based on the medication "ON" state and the "ON" UPDRS III score was limited with respect to reflecting the severity of PD patient motor symptoms [34]. In contrast, LEDD, which is a surrogate marker of disease progression, was correlated with K-COMPASS-31 total and subscores [15]. In addition, K-COMPASS-31 score was correlated with disease duration, NMSS, and PDQ-39 SI in PD patients, and these results coincided with a previous study that suggested an association between longer disease duration and severe autonomic dysfunction [1].

Table 3. Correlation among K-COMPASS-31 scores, clinical features, and the results of autonomic function tests.

| K-COMPASS-31 domains | Age[a] | Disease duration[b] | SCOPA-AUT[c] | AFT | | | UPDRS part III | | | | | H&Y[c] | MMSE[c] | LEDD[e] | NMSS[c] | PDQ-39-SI[c] |
|---|---|---|---|---|---|---|---|---|---|---|---|---|---|---|---|---|
| | | | | E:I ratio[c] | Valsalva ratio[c] | PRT[c] | Total[d] | Axial[d] | Tremor[d] | Rigidity[d] | Bradykinesia[d] | | | | | |
| Total score | -0.045 | 0.328[g] | 0.609[g] | -0.261[g] | -0.187 | 0.243[g] | 0.194 | 0.223[c] | -0.038 | 0.215[g] | 0.150 | 0.076 | -0.197 | 0.117 | 0.370[g] | 0.368[g] |
| Orthostatic intolerance | 0.046 | 0.227[g] | 0.472[g] | -0.163 | -0.136 | 0.220[g] | 0.135 | 0.099[g] | 0.004 | 0.160 | 0.124 | 0.037 | -0.187 | 0.132 | 0.306[g] | 0.300[g] |
| Vasomotor | 0.058 | 0.237[g] | 0.240[g] | -0.145 | -0.237[g] | 0.248[g] | 0.040 | -0.061 | 0.088 | 0.049 | 0.051 | -0.126 | -0.048 | 0.080 | 0.108 | 0.113 |
| Secretomotor | -0.057 | 0.291[g] | 0.456[g] | -0.194 | -0.203 | 0.140 | 0.409[g] | 0.446[g] | -0.013 | 0.378[g] | 0.332[g] | 0.173 | -0.077 | -0.040 | 0.319[g] | 0.300[g] |
| Gastrointestinal | -0.226[g] | 0.192 | 0.559[g] | -0.344[g] | -0.181 | 0.217[g] | 0.060 | 0.180[g] | -0.084 | 0.060 | 0.007 | -0.007 | -0.169 | 0.089 | 0.174 | 0.208 |
| Bladder | -0.056 | 0.408[g] | 0.474[g] | -0.255[g] | -0.041 | 0.137 | -0.051 | 0.119 | -0.193 | 0.064 | -0.120 | 0.173 | -0.132 | 0.033 | 0.313[g] | 0.330[g] |
| Pupillomotor | -0236[g] | 0.286[g] | 0.121 | -0.016 | -0.0065 | -0.201 | 0.086 | 0.206 | -0.043 | 0.032 | 0.046 | 0.055 | -0.007 | 0.148 | 0.288[g] | 0.234 |

K-COMPASS-31, Korean version of the composite autonomic symptom score-31; SCOPA-AUT, Scale for outcomes in Parkinson's disease-autonomic; E:I, Expiratory:inspiratory; PRT, pressure recovery time; UPDRS, Unified Parkinson's disease rating scale; H&Y, Modified Hoehn and Yahr; MMSE, mini-mental state examination; LEDD, levodopa equivalent daily dose; NMSS, non-motor symptoms scale for Parkinson's disease; PDQ-39 SI, Parkinson's disease questionnaire-39 summary index.

[a]Pearson partial correlation test adjusted for sex, disease duration, education years, Parkinson's disease subtype, UPDRS III, and LEDD.

[b]Pearson partial correlation test adjusted for age, sex, education years, Parkinson's disease subtype, UPDRS III, and LEDD.

[c]Pearson partial correlation test adjusted for age, sex, disease duration, education years, Parkinson's disease subtype, UPDRS III, and LEDD.

[d]Pearson partial correlation test adjusted for age, sex, disease duration, education years, Parkinson's disease subtype, and LEDD.

[e]Pearson partial correlation test adjusted for age, sex, disease duration, education years, Parkinson's disease subtype, and UPDRS III.

[g]p value < 0.05.

The limitation of the present study is that most of the enrolled patients were taking PD medication, and evaluation was done during the medication 'ON' state. Dopaminergic medications can affect the presence or severity of autonomic symptoms and the results of objective testing, which might be a confounder in our study. However, because most patients with PD are treated daily with dopaminergic medications, evaluation of autonomic function in medicated patients might be a suitable comparison for real clinical practice.

In conclusion, the K-COMPASS-31 showed excellent reliability and validity for the assessment of autonomic symptoms in PD patients. The COMPASS-31 is easy to repeat and is widely used to investigate autonomic dysfunction in various neurologic disorders; therefore, its use in PD would allow comparison of autonomic dysfunction among different neurologic disorders. We believe that this questionnaire is valid to use in clinical practice for patients with PD and it is expected to be validated in various neurologic disorders.

## Supporting information

**S1 Data.**
(XLSX)

## Acknowledgments

## Declarations

**Disclosure.** J. Youn received speakers' honoraria from SK Chemicals, Boston Scientific, and research support from Medtronic and Boston Scientific.

**Ethics approval/Consent to participate.** This study was approved by the local institutional review board of each involved hospital, and all enrolled subjects provided written informed consent (SMC 2019-02-121).

## Author Contributions

**Conceptualization:** Jin Myoung Seok, Jinyoung Youn.

**Data curation:** Jong Hyeon Ahn, Jin Myoung Seok, Jongkyu Park, Heejeong Jeong, Younsoo Kim, Ju-Hong Min, Byoung Joon Kim, Jinyoung Youn.

**Formal analysis:** Jong Hyeon Ahn, Jin Myoung Seok, Jinyoung Youn.

**Investigation:** Jongkyu Park, Heejeong Jeong, Younsoo Kim, Joomee Song, Inyoung Choi, Jin Whan Cho, Ju-Hong Min.

**Methodology:** Jong Hyeon Ahn, Jin Myoung Seok, Jinyoung Youn.

**Project administration:** Byoung Joon Kim, Jinyoung Youn.

**Supervision:** Jin Whan Cho, Byoung Joon Kim, Jinyoung Youn.

**Writing – original draft:** Jong Hyeon Ahn, Jin Myoung Seok.

**Writing – review & editing:** Jong Hyeon Ahn, Jin Myoung Seok, Byoung Joon Kim, Jinyoung Youn.

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
