## [Decision Letter · Decision Letter 0]

8 Jun 2021

PONE-D-21-13501

Validation of the Korean version of the composite autonomic symptom scale 31 in patients with Parkinson’s disease

PLOS ONE

Dear Dr. Youn,

Thank you for submitting your manuscript to PLOS ONE. After careful consideration, we feel that it has merit but does not fully meet PLOS ONE’s publication criteria as it currently stands. Therefore, we invite you to submit a revised version of the manuscript that addresses the points raised during the review process.Please submit your revised manuscript by Jul 16 2021 11:59PM. If you will need more time than this to complete your revisions, please reply to this message or contact the journal office at plosone@plos.org. Please include the following items when submitting your revised manuscript:

We look forward to receiving your revised manuscript.

Kind regards,

Antonina Luca, MD, PhD

Academic Editor

PLOS ONE

Journal Requirements:

2.Thank you for including your ethics statement:  "This study was approved by the local Institutional Review Board of each involved hospital, and all enrolled subjects provided written informed consent (2019-02-121)"

3.We note that you have indicated that data from this study are available upon request. PLOS only allows data to be available upon request if there are legal or ethical restrictions on sharing data publicly. For information on unacceptable data access restrictions, please see http://journals.plos.org/plosone/s/data-availability#loc-unacceptable-data-access-restrictions.

5.Thank you for stating the following in the Acknowledgments Section of your manuscript:

"J. Youn declares speaker’s honoraria from SK Chemicals, Boston Scientific, and research

86 support from Medtronic and Boston Scientific.

87

88 The other authors have no conflicts of interest to disclose."

" The author(s) received no specific funding for this work."

Additionally, because some of your funding information pertains to commercial funding, we ask you to provide an updated Competing Interests statement, declaring all sources of commercial funding.

In your Competing Interests statement, please confirm that your commercial funding does not alter your adherence to PLOS ONE Editorial policies and criteria by including the following statement: "This does not alter our adherence to PLOS ONE policies on sharing data and materials.” as detailed online in our guide for authors  http://journals.plos.org/plosone/s/competing-interests.  If this statement is not true and your adherence to PLOS policies on sharing data and materials is altered, please explain how.

Please include the updated Competing Interests Statement and Funding Statement in your cover letter. We will change the online submission form on your behalf.

Reviewers' comments:

Reviewer's Responses to Questions

**Comments to the Author**

1. Is the manuscript technically sound, and do the data support the conclusions?

Reviewer #1: Partly

Reviewer #2: Yes

2. Has the statistical analysis been performed appropriately and rigorously? 

Reviewer #1: No

Reviewer #2: Yes

3. Have the authors made all data underlying the findings in their manuscript fully available?

Reviewer #1: No

Reviewer #2: No

4. Is the manuscript presented in an intelligible fashion and written in standard English?

Reviewer #1: Yes

Reviewer #2: Yes

5. Review Comments to the Author

Reviewer #1: Validation of the Korean version of the composite autonomic symptom scale 31 in patients with Parkinson’s disease

In the present study the Authors developed a Korean version of COMPASS-31 and verified its reliability and validity in patients with PD. The study design is appropriate, the sample size is adequate, the paper is well-written although a review by an experienced English speaker is recommended. Nevertheless, I have some major points of criticism:

1)Materials and methods:

-Subjects and clinical assessment:

•the Authors assessed possible correlation between K-COMPASS-31 score and Axial, Tremor, Rigidity and Bradykinesia UPDRS III subscores. In this section a formal definition of these UPDRS III subscores is needed.

•Clinical PD phenotypes should be described in study population using a standardized approach and taken into account in data interpretation (Stebbins GT et al. How to identify tremor dominant and postural instability/gait difficulty groups with the movement disorder society unified Parkinson's disease rating scale: comparison with the unified Parkinson's disease rating scale. Mov Disord. 2013 May;28(5):668-70). Otherwise, the lack of PD phenotypes characterization must be considered in interpreting the results and therefore adequately discussed.

-Statistical analysis:

•an internal consistency analysis has been appropriately performed by using Cronbach’s α-coefficient. However, it is not clear if a factorial analysis has been performed in order to assess the internal validity of K-COMPASS-31 domains’ scores. In fact, even if it is not described in “Statistical analysis” section, in “Results” (Line 220) the Authors state that all of the K-COMPASS-31 domains showed good internal validity. Therefore, this point must be clarified and Cronbach’s α-coefficients for each domain adequately reported.

•Considering the large number of PD patients enrolled in the study, a Pearson’s correlation analysis could be performed.

•The external validity of K-COMPASS-31 has been properly assessed by a correlation analysis with AFT and SCOPA-AUT. In particular, a correlation analysis between K-COMPASS-31 total score and subscores and SCOPA-AUT total score has been performed. I believe that could be useful to explore potential correlation between corresponding subscores of both scales, while it is not useful to assess a correlation between K-COMPASS-31 subscores and SCOPA-AUT total score.

•Correlation analysis has been appropriately adjusted by potential confounders. In line 187 the Authors state that potential confounders included: age, disease duration, education years and UPDRS III. Nevertheless, according to results (Table 3), disease duration does not seem to be considered in adjusted correlation analysis. In particular, I believe that disease duration (which was significantly correlated with all of the K-COMPASS-31 score domains) should be considered as a potential confounder in correlation analysis between K-COMPASS-31 and all of the variables assessed. Similarly, LEDD should be considered in correlations with clinical severity-related scores. Finally, also sex should be considered as a priori potential confounder.

2)Results and Discussion: a statistically significant age-adjusted positive correlation between K-COMPASS-31 (total score and orthostatic intolerance, secretomotor and gastrointestinal subscores) and PRT has been found. Considering that PRT indicates an adrenergic failure, the Authors suggest that K-COMPASS-31 may be useful in evaluating the characteristics of orthostatic hypotension in PD. This point should be better discussed considering that the AFT battery used did not encompass the Active Standing test which conceptually differs from HUT. Moreover, HUT test results must be shown and properly discussed.

3)Tables:

-Table 3: superscript letters do not match table description. Moreover, a more definite separation between UPDRS-III and AFT description is needed.

Reviewer #2: The manuscript is technically sound piece of scientific research with data that supports the conclusions. the manuscript presented in an intelligible fashion. Also statistical analysis was appropriate. But there are restrictions about data availability.

For Authors: would you explain the cut off point of the Korean version of the composite autonomic symptom scale 31 to differentiate PD patients with and without autonomic disturbance, and the sensitivity and specificity of it.

6. PLOS authors have the option to publish the peer review history of their article (what does this mean?). If published, this will include your full peer review and any attached files.

Reviewer #1: No

Reviewer #2: No

---

## [Author Response · Author response to Decision Letter 0]

15 Sep 2021

PONE-D-21-13501

Title: Validation of the Korean version of the composite autonomic symptom scale 31 in patients with Parkinson’s disease

Dear Antonina Luca, MD, PhD Academic Editor at PLoS One,

We would like to express our sincere gratitude for your thorough consideration and scrutiny of our manuscript. Through the accurate and keen comments made by the reviewers, we realized the critical points at issue in our analyses and manuscript. After we received the reviewers’ criticisms, we exerted our best effort to achieve the scientific and literary levels required by PLoS One.

This response and the revised manuscript are the result of our hard work to respond to the comments. Additionally, we revised our manuscript as a short communication, thus there are a few more changes other than the response to reviewers’ comments. We hope our revised manuscript will be considered positively and be accepted by PLoS One.

 

Reviewer #1: 

In the present study the Authors developed a Korean version of COMPASS-31 and verified its reliability and validity in patients with PD. The study design is appropriate, the sample size is adequate, the paper is well-written although a review by an experienced English speaker is recommended. Nevertheless, I have some major points of criticism:

Response: Thank you for giving us the opportunity to strengthen our manuscript with your valuable comments and queries. Our manuscript has been revised by professional English editing service (eWorldEditing, https://www.eworldediting.com/).

1)Materials and methods:

-Subjects and clinical assessment:

•the Authors assessed possible correlation between K-COMPASS-31 score and Axial, Tremor, Rigidity and Bradykinesia UPDRS III subscores. In this section a formal definition of these UPDRS III subscores is needed.

Response: Thank you for the comment. We have added the definition of the UPDRS subscores and the reference at the methods section. (Page 8, Line 150-153).

Page 8, Line 150-153

The UPDRS part III was grouped into tremor (items 20 and 21), rigidity (item 22), bradykinesia (items 23, 24, 25, 26, and 31), and axial motor symptoms (items 27, 28, 29, and 30) [13].

•Clinical PD phenotypes should be described in study population using a standardized approach and taken into account in data interpretation (Stebbins GT et al. How to identify tremor dominant and postural instability/gait difficulty groups with the movement disorder society unified Parkinson's disease rating scale: comparison with the unified Parkinson's disease rating scale. Mov Disord. 2013 May;28(5):668-70). Otherwise, the lack of PD phenotypes characterization must be considered in interpreting the results and therefore adequately discussed.

Response: Thanks for the valuable comment. As you pointed out, the motor subtypes of PD could be associated with the autonomic involvements. Two classifications for motor subtypes were widely used; one is tremor dominant (TD) vs. postural instability/gait difficulty (PIGD), and the other is TD vs. akinetic-rigid (AR) subtype [1]. In general, the non-TD (PIGD or AR) subtypes seem to show more advanced neurodegeneration and less favorable outcomes than the TD subtype. In this study, we evaluated parkinsonism with Unified Parkinson’s disease rating scale (UPDRS), not the movement disorder society (MDS) UPDRS, and used UPDRS III without UPDRS II. Therefore, we have divided the subjects into TD, AR, and mixed subtypes (table 1), and we hope that this is acceptable. 

When we compared the clinical and demographic data among the subtypes, there were no significant differences except that AR subtype showed the higher LEDD (the first table attached on the response to reviewers file). In terms of the K-COMPASS-31, AR subtype had higher secretomotor domain score than TD, but there were no differences in other domains (the second table attached on the response to reviews file). We thought the differences from motor subtypes could be biased because we examined the enrolled subjects on ‘ON’ state. Considering the previous study [1] that 72.2% of participants were evaluated in the off state, the subtype in our study could be changed with medication. Additionally, we tried to validate Korean version of COMPASS-31 in PD patients in this study, thus we added the subtypes of enrolled subjects in table 1, but we did not show the comparison among the subtypes in our manuscript. For the correlation analysis, we definitely agree with your opinion that motor subtype is important factor, thus we added motor subtypes as the confounder factor for the correlation analysis. However, if you still think it is better to show the data among subtypes in table 1, we are happy to add the additional data.

Methods (Page 8, line 152-153)

Patients were classified into three subtypes according to dominant parkinsonian symptoms: tremor-dominant (TD), akinetic-rigid (AR) and the mixed subtypes [14].

Reference

[1] Kang GA, Bronstein JM, Masterman DL, Redelings M, Crum JA, Ritz B. Clinical characteristics in early Parkinson's disease in a central California population-based study. Mov Disord. 2005 Sep;20(9):1133-42.

Statistical analysis:

•an internal consistency analysis has been appropriately performed by using Cronbach’s α-coefficient. However, it is not clear if a factorial analysis has been performed in order to assess the internal validity of K-COMPASS-31 domains’ scores. In fact, even if it is not described in “Statistical analysis” section, in “Results” (Line 220) the Authors state that all of the K-COMPASS-31 domains showed good internal validity. Therefore, this point must be clarified and Cronbach’s α-coefficients for each domain adequately reported.

Response: The reviewer’s point was well-taken. We have added Cronbach’s α-coefficient in each domains and Cronbach’s α-coefficient of the K-COMPASS 31 subdomains showed similar with the original study. We have evaluated the validity of the K-COMPASS-31 using Pearson’s partial correlation analysis with SCOPA-AUT, and the K-COMPASS-31 showed well correlated with the SCOPA-AUT. The results have been presented in Table 3 and Figure 1 (F) of the manuscript (Page 10, line 191-198 and Page 13, line 248-252).

Methods (Page 10, line 191-198)

Cronbach’s α-coefficient was used for internal consistency analysis of the K-COMPASS-31 total and for each domain, and test-retest reliability was assessed using the intra-class correlation coefficient (ICC). Validity was assessed using correlation analysis between the K-COMPASS-31 and SCOPA-AUT; Pearson partial correlation was adjusted for potential confounders that included age, sex, disease duration, education year, PD subtype UPDRS III, and LEDD. The association of the K-COMPASS-31 with other variables including age, disease duration, UPDRS, H&Y, LEDD, NMSS, and PDQ-39 summary index (PDQ-39 SI) was also investigated.

Results (Page 13, line 248-252)

K-COMPASS-31 total scores were positively correlated with SCOPA-AUT scores (r = 0.626, p < 0.001). Subdomain scores of the K-COMPASS-31 were also correlated with SCOPA-AUT total score. The pupillomotor domain of the K-COMPASS-31 was not correlated with total SCOPA-AUT, but was correlated with the pupillomotor subscore of SCOPA-AUT (Fig 1).

•Considering the large number of PD patients enrolled in the study, a Pearson’s correlation analysis could be performed.

Response: Thank you for the comment. We have performed the Pearson’s correlation analysis rather than Spearman correlation analysis and replace the methods and results as you recommended (page, line ,table 3 and Figure 1)

Methods (page, line)

Validity was assessed by the correlation analysis between the K-COMPASS-31 and SCOPA-AUT, using Pearson partial correlation adjusting for potential confounders that included age, sex, disease duration, education year, PD subtype UPDRS III, and LEDD. The association of the K-COMPASS-31 and other variables including age, disease duration, UPDRS, H&Y, LEDD, NMSS, and PDQ-39 summary index (PDQ-39 SI) were investigated, as well.

•The external validity of K-COMPASS-31 has been properly assessed by a correlation analysis with AFT and SCOPA-AUT. In particular, a correlation analysis between K-COMPASS-31 total score and subscores and SCOPA-AUT total score has been performed. I believe that could be useful to explore potential correlation between corresponding subscores of both scales, while it is not useful to assess a correlation between K-COMPASS-31 subscores and SCOPA-AUT total score.

Response: Thanks for the insightful comment. As you suggested, we have performed correlation analysis between K-COMPASS-31 (total and subscores) and SCOPA-AUT (total and subscores) as you can find out in the figure 1 (F) (correlation matrix). The K-COMPASS-31 scores showed correlation with the SCOPA-AUT total and subscore. Vasomotor domain of the K-COMPASS-31 and pupillomotor score did not show significant correlation with other symptoms, but they were well correlated with the associated SCOPA-AUT subscores (Vasomotor of COMPASS and cardiovascular, pupillomotor of COMPASS-31 and SCOPA-AUT). We had also added brief discussion about the correlations (Page Line).

Discussion (Page Line)

 However, the K-COMPASS-31, and its subscores were significantly correlated with both the SCOPA-AUT and the objective parameters of AFT in our study.

•Correlation analysis has been appropriately adjusted by potential confounders. In line 187 the Authors state that potential confounders included: age, disease duration, education years and UPDRS III. Nevertheless, according to results (Table 3), disease duration does not seem to be considered in adjusted correlation analysis. In particular, I believe that disease duration (which was significantly correlated with all of the K-COMPASS-31 score domains) should be considered as a potential confounder in correlation analysis between K-COMPASS-31 and all of the variables assessed. Similarly, LEDD should be considered in correlations with clinical severity-related scores. Finally, also sex should be considered as a priori potential confounder.

Response: Your point was well-taken. As you suggested, we have performed the Pearson partial correlation with potential confounders using, age, sex, disease duration, LEDD, UDPRS III, and educational years. 

There was no difference between re-analysis and original analysis, except the correlation between the Valsalva ratio and the K-COMPASS-31 total score. The results suggest that E:I ratio might be more suitable for evaluation of cardiovagal dysfunction in patients with Parkinson’s disease because of difficulties in the performance of Valsalva maneuver. We have revised the methods, results and discussion section according to the re-analysis ((page10, line 193-198, page 13, line 240-252, and page 17, line 331-337, Table 3, Figure 1).

Methods (page10, line 193-198)

Validity was assessed using correlation analysis between the K-COMPASS-31 and SCOPA-AUT; Pearson partial correlation was adjusted for potential confounders that included age, sex, disease duration, education year, PD subtype UPDRS III, and LEDD. The association of the K-COMPASS-31 with other variables including age, disease duration, UPDRS, H&Y, LEDD, NMSS, and PDQ-39 summary index (PDQ-39 SI) was also investigated.

Results (page 13, line 240-252)

There were significant correlations between K-COMPASS-31 score and the AFT results. E:I ratio and Valsalva ratio were negatively correlated (r = -0.240, p = 0.023; r = -0.247, p = 0.019) indicating cardiovagal dysfunction [23], and pressure recovery time (PRT), which represented adrenergic function, was positively correlated with K-COMPASS-31 score (r = 0.345, p < 0.001) (Fig 1). However, after adjusting for confounders including age, sex, disease duration, education year, PD subtype UPDRS III, and LEDD, K-COMPASS-31 score was correlated with E:I ratio and PRT (Table 3). 

 K-COMPASS-31 total scores were positively correlated with SCOPA-AUT scores (r = 0.626, p < 0.001). Subdomain scores of the K-COMPASS-31 were also correlated with SCOPA-AUT total score. The pupillomotor domain of the K-COMPASS-31 was not correlated with total SCOPA-AUT, but was correlated with the pupillomotor subscore of SCOPA-AUT (Fig 1).

Discussion (Page 17, line 331-337)

For the evaluation of cardiovagal dysfunction, K-COMPASS-31 showed a negative correlation with E:I ratio, but was not correlated with Valsalva ratio after adjusting for confounders. A previous validation study of the COMPASS-31 showed similar results, in that COMPASS-31 correlated well with overall AFT score but not with AFT scores for cardiovagal function [26]. E:I ratio might be more suitable for evaluating cardiovagal dysfunction in patients with Parkinson’s disease because of difficulties in performing the Valsalva manoeuvre [33].

2)Results and Discussion: a statistically significant age-adjusted positive correlation between K-COMPASS-31 (total score and orthostatic intolerance, secretomotor and gastrointestinal subscores) and PRT has been found. Considering that PRT indicates an adrenergic failure, the Authors suggest that K-COMPASS-31 may be useful in evaluating the characteristics of orthostatic hypotension in PD. This point should be better discussed considering that the AFT battery used did not encompass the Active Standing test which conceptually differs from HUT. Moreover, HUT test results must be shown and properly discussed.

Response: Thank you for this comment. The active standing test (AST) was not included in our AFT battery. Considering the testing methods of AST and HUT, AST is more physiologically feasible test mimics daily activities. However, older adults with parkinsonism cannot change their position easily, HUT might be more convenient and suitable method to detect orthostatic hypotension (OH) [1] both tests are widely accepted and used as diagnostic tools for evaluation of orthostatic intolerance [2]. In our study, 23 patients (25.6%) showed OH during HUT, which is assigned according to the previous consensus statement on the definition of OH [3]. The scores of K-COMPASS-31 was higher in patients with OH than those without OH (33.4±21.4 vs. 18.1 ±13.9, p < 0.001); orthostatic intolerance subscore was also higher in patients with OH (16.5±13.1 vs. 7.3±9.6, p = 0.001). Therefore, K-COMPASS-31 score might be helpful to evaluate OH in patients with PD. We have revised Table 1 and Figure 1 and discussion section. 

References

1. Aydin AE, Soysal P, Isik AT. Which is preferable for orthostatic hypotension diagnosis in older adults: active standing test or head-up tilt table test? Clin Interv Aging 2017;12:207-212.

2. Brignole M, Moya A, de Lange FJ, et al. 2018 ESC Guidelines for the diagnosis and management of syncope. Eur Heart J 2018;39:1883-1948.

3. Freeman R, Wieling W, Axelrod FB, et al. Consensus statement on the definition of orthostatic hypotension, neurally mediated syncope and the postural tachycardia syndrome. Clin Auton Res 2011;21:69-72.

Discussion (Page 16, line 321-331)

In this study, 23 patients (25.6%) showed OH during HUT. The patients with OH had a higher total K-COMPASS-31 score and orthostatic intolerance subscore than those without OH (33.4 ± 21.4 vs. 18.1 ± 13.9, p < 0.001; 16.5±13.1 vs. 7.3 ± 9.6, p = 0.001). The total score and orthostatic subscore of the K-COMPASS-31 correlated moderately with PRT (r = 0.243; r = 0.220), which reflects adrenergic dysfunction and is defined as the time interval from the time of lowest blood pressure in phase 3 to when the blood pressure reaches baseline during the Valsalva manoeuvre [23, 32]. Considering this difference in K-COMPASS-31 scores according to the presence of OH and the correlation between PRT and the K-COMPASS-31 scores, the K-COMPASS-31 score, especially its orthostatic intolerance subscore, might be helpful for evaluating characteristics of OH in patients with PD.

3)Tables:

-Table 3: superscript letters do not match table description. Moreover, a more definite separation between UPDRS-III and AFT description is needed.

Response: Thank you for the comment. We have modified the table 3 to make easier to understand, as you recommended.

Reviewer #2: 

The manuscript is technically sound piece of scientific research with data that supports the conclusions. the manuscript presented in an intelligible fashion. Also statistical analysis was appropriate. But there are restrictions about data availability.

Response: Thank you for the comment. We can provide all the anonymized data under the approval of IRB if there’s reasonable request for raw data. We have added the data availability statement at the manuscript (page 5, line 84)

Data availability statement (Page 5, line 84)

The full data are not publicly available because of participant privacy and consent. Anonymized data will be shared by upon reasonable requests from any qualified investigators for 3 years after the date of publication.

For Authors: would you explain the cutoff point of the Korean version of the composite autonomic symptom scale 31 to differentiate PD patients with and without autonomic disturbance, and the sensitivity and specificity of it.

Response: Thank you for the valuable comment. It is great idea to set cut-off value for the presence of autonomic dysfunction. However, autonomic involvements in PD can present with diverse symptoms and it is difficult to dichotomize PD patients based on the present of autonomic dysfunction. In PD, most of them have some degree of dysautonomic symptoms, and there were no definition or gold standard to define the dysautonomia in PD. In the present study, only 4 of 90 (4.4%) patients showed ‘0’ score in the K-COMPASS-31, cutoff value of the K-COMPASS-31 not that useful for the PD. However, the COMPASS-31 would be useful for the differential diagnosis of PD from MSA-P, as we investigated before [1], and we hope further investigations will be conducted using the K-COMAPSS-31 for comparing the autonomic dysfunction in various disorders. 

Reference

1. Kim Y, Seok JM, Park J, Kim KH, Min JH, Cho JW, Park S, Kim HJ, Kim BJ, Youn J. The composite autonomic symptom scale 31 is a useful screening tool for patients with Parkinsonism. PLoS One. 2017 Jul 6;12(7):e0180744.

---

## [Decision Letter · Decision Letter 1]

8 Oct 2021

Validation of the Korean version of the composite autonomic symptom scale 31 in patients with Parkinson’s disease

PONE-D-21-13501R1

Dear Dr. Jinyoung Youn

We’re pleased to inform you that your manuscript has been judged scientifically suitable for publication and will be formally accepted for publication once it meets all outstanding technical requirements.

Kind regards,

Antonina Luca, MD, PhD

Academic Editor

PLOS ONE

Reviewers' comments:

Reviewer's Responses to Questions

**Comments to the Author**

1. If the authors have adequately addressed your comments raised in a previous round of review and you feel that this manuscript is now acceptable for publication, you may indicate that here to bypass the “Comments to the Author” section, enter your conflict of interest statement in the “Confidential to Editor” section, and submit your "Accept" recommendation.

Reviewer #1: All comments have been addressed

Reviewer #2: All comments have been addressed

2. Is the manuscript technically sound, and do the data support the conclusions?

Reviewer #1: (No Response)

Reviewer #2: Yes

3. Has the statistical analysis been performed appropriately and rigorously? 

Reviewer #1: (No Response)

Reviewer #2: Yes

4. Have the authors made all data underlying the findings in their manuscript fully available?

Reviewer #1: (No Response)

Reviewer #2: (No Response)

5. Is the manuscript presented in an intelligible fashion and written in standard English?

Reviewer #1: (No Response)

Reviewer #2: (No Response)

6. Review Comments to the Author

Reviewer #1: All comments have been adequately addressed. I only suggest to indicate a reference for lines 319-322.

Reviewer #2: (No Response)

7. PLOS authors have the option to publish the peer review history of their article (what does this mean?). If published, this will include your full peer review and any attached files.

Reviewer #1: No

Reviewer #2: No

---

## [Editor Report · Acceptance letter]

13 Oct 2021

PONE-D-21-13501R1 

Validation of the Korean version of the composite autonomic symptom scale 31 in patients with Parkinson’s disease 

Dear Dr. Youn:

I'm pleased to inform you that your manuscript has been deemed suitable for publication in PLOS ONE. Congratulations! Your manuscript is now with our production department. 

Kind regards, 

on behalf of

Dr. Antonina Luca 

Academic Editor

PLOS ONE